# Light-induced MOF synthesis enabling composite photothermal materials

Ofir Shelonchik [1], Nir Lemcoff [1], Ran Shimoni[1], Aritra Biswas [1], Elad Yehezkel[1], Doron Yesodi[1], Idan Hod[1,2,3] & Yossi Weizmann [1,2,3] ✉

Metal-organic frameworks (MOFs) are a class of porous materials known for their large surface areas. Thus, over the past few decades the development of MOFs and their applications has been a major topic of interest throughout the scientific community. However, many current conventional syntheses of MOFs are lengthy solvothermal processes carried out at elevated temperatures. Herein, we developed a rapid light-induced synthesis of MOFs by harnessing the plasmonic photothermal abilities of bipyramidal gold nanoparticles (AuBPs). The generality of the photo-induced method was demonstrated by synthesizing four different MOFs utilizing three different wavelengths (520 nm, 660 nm and 850 nm). Furthermore, by regulating light exposure, AuBPs could be embedded in the MOF or maintained in the supernatant. Notably, the AuBPs-embedded MOF (AuBP@UIO-66) retained its plasmonic properties along with the extraordinary surface area typical to MOFs. The photothermal AuBP@UIO-66 demonstrated a significant light-induced heating response that was utilized for ultrafast desorption and MOF activation.

Metal-organic frameworks (MOFs) are materials that consist of metal ions or clusters connected by organic linkers to form crystalline structures[1]. This unique arrangement results in highly porous materials with enormous surface areas[2] that can be tailored for exact pore size and functionality by utilizing suitable ions/clusters and linkers[3]. The remarkable properties and versatility demonstrated by MOFs have led to extensive research towards developing applications in various fields, such as gas storage and separation[4,5], drug delivery[6], sensing[7], catalysis[8,9], electronic devices[10], and membranes[11]. Thus, the synthesis of MOFs has become an area of broad interest and relevance. However, many of today's benchmark synthesis procedures are carried out via time-consuming solvothermal methods[3]. This synthetic approach suffers from being energetically inefficient, a crucial disadvantage for large-scale production of MOFs, and a major setback in their transition from the lab to industry[12]. Consequently, much attention has been devoted to developing new, efficient routes for MOF formation[13–15]. Some noteworthy examples are microwave-assisted[16], electrochemical[17], and mechanical syntheses[18]. Although

these strategies offer different potential advantages, there is still a need to improve on current procedures and provide effective alternative MOF syntheses[3]. To the best of our knowledge, this work represents the first example of a visible (Vis) or near-infrared (NIR) light-induced MOF synthesis achieved by utilizing photothermal materials.

Plasmonic nanoparticles (PNPs) can convert light to heat via the "localized surface plasmon resonance" (LSPR) effect. The interaction between an incoming electromagnetic wave at the resonant frequency and the nanoparticle's conduction electrons initiates a cascade of events, resulting in heat dissipation from the PNP[19]. The resonant frequency is highly dependent on the exact PNP geometry, making the heating efficiency sensitive to subtle changes in the PNP structure. The plasmonic effect has been extensively researched over the past few decades and has been introduced into various fields such as biosensing[20], spectroscopic applications[21], and photothermal cancer therapy[22]. From the plethora of plasmonic materials developed so far[23], gold nanoparticles are the most popular choice for application as

[1]Department of Chemistry, Ben-Gurion University of the Negev, Beer-Sheva 84105, Israel. [2]Ilse Katz Institute for Nanotechnology Science, Ben-Gurion University of the Negev, Beer-Sheva 84105, Israel. [3]Goldman Sonnenfeldt School of Sustainability and Climate Change, Ben-Gurion University of the Negev, Beer-Sheva 84105, Israel. ✉e-mail: yweizmann@bgu.ac.il

photothermal converters[24]. Their excellent chemical stability and biocompatibility[25] enable them to function in versatile environments, and their exceptional plasmonic properties[26] lead to highly efficient light-to-heat conversion. Moreover, a variety of different gold nanostructures have well-defined syntheses, thus facilitating the possibility of manipulating the photothermal activation wavelength by changing the nanoparticle size and shape[27]. Gold nano bipyramids (AuBPs) are an ideal example since their synthesis yields highly monodisperse structures, and they can be tuned to afford different-sized AuBPs[28,29]. Furthermore, the unique bipyramidal shape has an intrinsic relatively narrow LSPR absorption band[30] and remarkable conversion efficiency[31]. These factors establish AuBPs as prominent tunable light-to-heat nanoconverters[32,33].

Herein, we report the light-induced synthesis of MOFs in the Vis and NIR regions by harnessing the photothermal capabilities of different nanomaterials. The generality of our method was demonstrated by synthesizing four types of MOFs (UIO-66, MIL-88A, HKUST-1, and MOF-5) utilizing different photothermal agents, including two sizes of AuBPs corresponding to two different photothermal activation wavelengths (660 and 850 nm), as well as gold nanospheres (520 nm activation wavelength) and carbon-based materials. Importantly, we discovered that the photothermal synthesis is rapid when compared with conventional methods; thus, it substantially improves the efficiency of the reaction.

## Results and discussion
### Light-induced MOF synthesis
To explore the possibility of developing a light-induced MOF synthesis, we modified the conventional solvothermal synthesis of the well-known UIO-66[34] reported by Katz et al.[35]. This zirconium-based MOF was first synthesized by J.Cavka et al. in 2008[36] and has since gained vast popularity for its exceptional aqueous, acidic, thermal, and mechanical robustness[37]. To initiate the synthesis of UIO-66, a dimethylformamide (DMF) solution that included terephthalic acid (BDC), $ZrCl_4$, and HCl was prepared, as suggested by Katz and co-workers. To this mixture, core–shell silica-coated AuBPs (AuBP@$SiO_2$) with a photothermal activation wavelength of 850 nm (AuBP$_{850}$, see Supplementary Fig. 1 for characterization) were added. AuBPs were encapsulated to avoid deterioration of the bipyramidal structure, causing a blue shift of the wavelength[38,39]. We then irradiated the reaction using a simple 8 W, 850 nm LED, and to our surprise, we noticed MOF formation within minutes of IR exposure (Supplementary Fig. 9). Motivated by the initial results, we prepared a series of solutions with varying AuBP concentrations and irradiated them for 30 min (Fig. 1b, c). The temperature profiles suggest that 0.5 OD of AuBP$_{850}$ is sufficient to sustain the temperature above 90 °C throughout the reaction while increasing the concentration to 2 OD led to elevated temperatures above 120 °C (Fig. 1b). Reaction yields were evaluated and compared to conventional overnight synthesis at 100 °C (Fig. 1c). Notably,

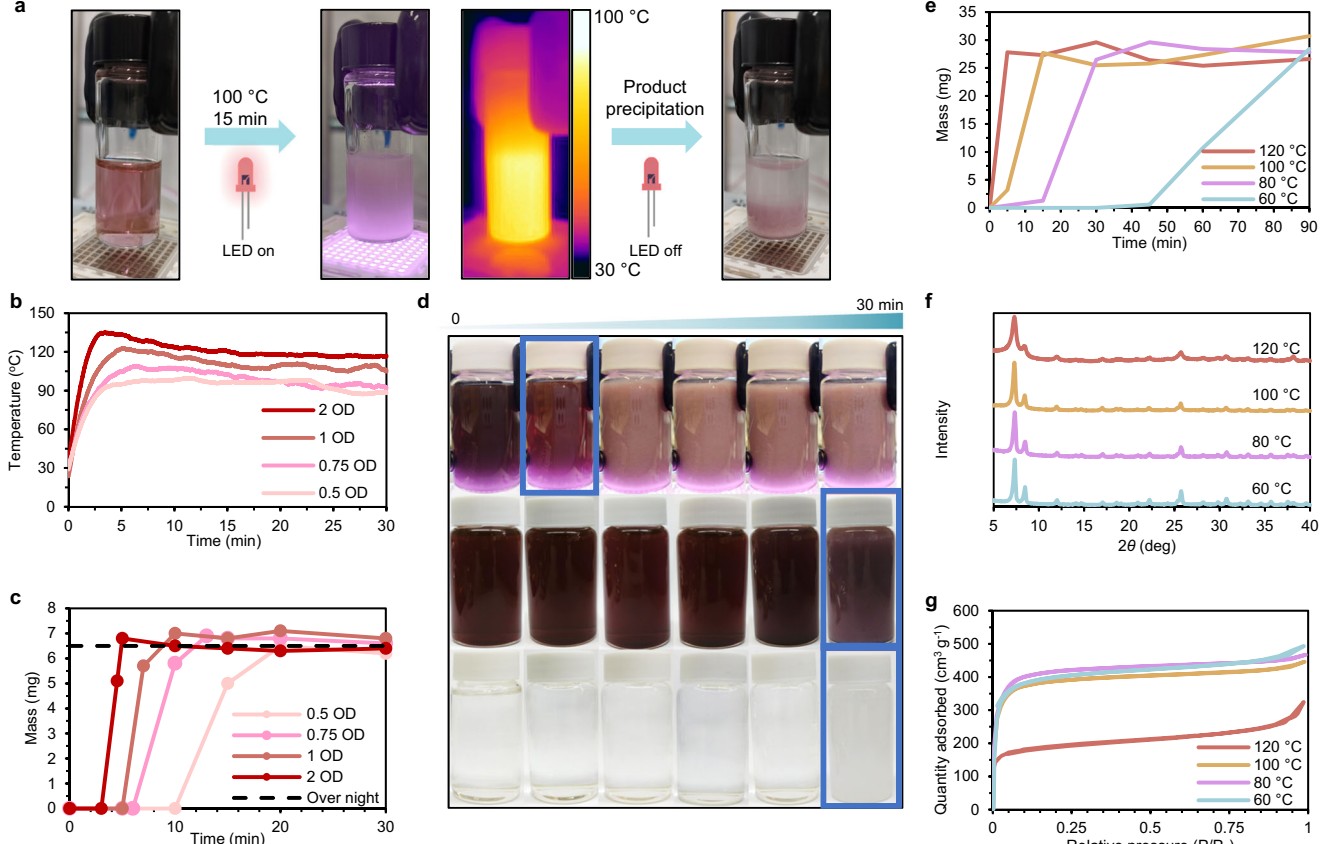

**Fig. 1 | Photothermal synthesis of UiO-66. a** Scheme of UIO-66 synthesis, in the volume of 2 mL, utilizing an 850 nm, 100 W LED. **b** Temperature profiles of UIO-66 photoinduced syntheses with different concentrations of AuBP$_{850}$. AuBP$_{850}$ concentration is defined via optical density (OD). **c** UIO-66 mass as a function of time for PPRs in (**a**). **d** Images showing the progress of 20 mL syntheses at 100 °C. Initial MOF formation is denoted by blue squares. The left column represents 0 min of reaction, then right to it 10 min, and then intervals of 5 min up to the column furthest to the left representing 30 min of reaction. Upper row: photothermal synthesis. Middle row: a conventional synthesis with AuBPs. Bottom row: conventional synthesis without AuBPs. The vials have a length of 6 cm and radius of 1.75 cm. **e** Product mass as a function of time for PPRs at different temperatures. **f** PXRD results of products from PPRs in (**e**). **g** BET $N_2$ adsorption isotherms of products from PPRs in (**e**).

the results indicated that all four photothermal reactions achieved the overnight yield within 20 min, even at lower temperatures.

To study the newfound rate improvement, we increased the volume of the reaction mixture (total of 20 mL with 5 OD AuBP$_{850}$) and compared the NIR-induced synthesis to the reactions that were carried out using a heating block with and without AuBPs (Fig. 1d). In the photothermal reaction, MOF formation could be observed within 10 min. In contrast, in both conventionally heated mixtures, MOF formation only started after 30 min, indicating that the presence of AuBPs alone does not contribute to the rate increase. Multiple studies on plasmonic photothermal nanoparticles show that the temperature near the NP's surface is considerably higher than the ambient temperature[40,41]. Therefore, we hypothesized that the AuBPs and their immediate surroundings act as hotspots where the temperature is greater than that of the bulk solution, causing the reaction to accelerate as if it were carried out in a hotter environment. Furthermore, the successful scaling of the process to a 20 mL volume, without compromising the time advantage of the photothermal synthesis, indicates its suitability for rapid, large-scale production. With the incorporation of suitable equipment, such as high-capacity LEDs, this process holds promise for industrial-scale implementation, offering faster results and greater energy efficiency.

Next, we studied the plasmonic photothermal reaction (PPR) at different temperatures by regulating the IR exposure (see supplementary Note 4 for the temperature regulation method). The results suggest that the same trend observed previously (Fig. 1c) occurred again, where higher temperatures resulted in quicker reactions (Fig. 1e). To ensure that the product quality is not affected when carrying out PPRs, we analyzed the product's powder X-ray diffractograms (PXRD) and the Brunauer–Emmett–Teller (BET) surface areas ($S_{BET}$) (Fig. 1f, g). The PXRD measurements confirmed that the resulting MOFs are identical to the UIO-66 reported in the literature. In addition, the $S_{BET}$ results indicate that MOFs synthesized via PPRs have greater surface areas than the conventionally synthesized UIO-66, with the exception of the PPR at 120 °C (Supplementary Table 5). The increased $S_{BET}$ value might be attributed to more defects in the MOF structure[42], resulting from the high temperatures in proximity to the AuBPs. Regarding the PPR carried out at 120 °C, the $S_{BET}$ value was substantially decreased, indicating that the UIO-66 structure was compromised beyond a point where it could be beneficial[43–45]. The fact that a decreased surface area was observed at 120 °C, contrary to data in reports utilizing conventional methods[35], strengthens the possibility for elevated local temperatures. To directly test this hypothesis, we conducted a photothermal synthesis of UIO-66 using Methylene blue, an organic dye that absorbs light at 660 nm. Methylene blue's molecular nature prevents the steep temperature gradients created between a nanoparticle surface and the solution, enabling us to assess whether they play a role in the rate increase. The results show that the photothermal synthesis of UIO-66 with Methylene blue at 60 °C took 4 h to complete, which corresponds to the duration of the synthesis under conventional heating methods (Supplementary Fig. 47). This experiment adds further support to the idea that the high temperatures generated in the close surroundings of the plasmonic AuBPs are a crucial factor in the acceleration of the MOF formation reaction.

### Embedding AuBPs in MOF via photothermal synthesis

While exploring the photothermal formation of UIO-66, we noticed that the reaction temperature had an effect on whether the AuBPs precipitated with the product or remained dispersed in the supernatant. Thus, to better understand the observed effect, samples and supernatants produced from PPRs at different temperatures were analyzed via absorption spectra, elemental analysis, and electron microscopy. The absorption spectra pointed to a clear trend: at lower temperatures, more AuBPs are dispersed in the supernatant (Fig. 2f). SEM images and ICP-OES data showed a complementary trend where

more AuBPs were embedded in the MOF at higher temperatures (Fig. 2a–c, e). These results clearly point to the presence of a temperature-sensitive mechanism by which the AuBPs are embedded into the MOF.

In an attempt to shed light on this hidden mechanism, we investigated samples of AuBP@UIO-66 synthesized at different temperatures. We started by analyzing high-resolution TEM images of the composite synthesized at 100 °C and found that the AuBPs were partially covered with UIO-66 particles, hinting at some form of interaction between them (Supplementary Fig. 28). This led us to consider the possibility that one of UIO-66's precursors (BDC or zirconium chloride) could be reacting with the silica surface at elevated temperatures. To test this, we prepared two solutions identical to those used for the photothermal formation of UIO-66 but missing either BDC or ZrCl$_4$ (see the "Methods" section). Each solution was divided into three fractions and photothermally heated to a different temperature (60, 80, and 100 °C). The AuBPs isolated from the solutions containing only BDC were subjected to zeta potential measurements to check whether some change to surface properties could be detected. The results showed no significant difference from controls that had undergone heating in the same environment without BDC (Supplementary Fig. 34). As for the solutions containing ZrCl$_4$, a clear difference could be observed; as the solution was heated to a higher temperature it became more opaque to a point where in the solution heated to 100 °C a precipitate including the AuBPs was formed (Supplementary Fig. 35).

This interesting trend strongly suggests that zirconium could be attached to the silica shell at higher temperatures. To get an idea of whether this was the case, ICP-OES was performed on washed samples of AuBPs isolated from the heated solutions. The results validated our suspicion that the AuBPs produced from the solution heated to 100 °C contained a significantly larger zirconium to gold ratio than the other two samples (Supplementary Fig. 36). To gain a more detailed understanding of the zirconium silica interaction, we conducted EDS elemental mapping under STEM mode and X-ray photoelectron spectroscopy (XPS) analysis. Elemental mapping coupled with STEM images showed that at 100 °C a zirconium shell formed on the silica, at 80 °C small patches could be seen on the silica, and at 60 °C no significant differences were observed (Supplementary Figs. 38–40). XPS analysis of the AuBPs from the sample heated to 100 °C revealed the presence of zirconium oxide (Supplementary Fig. 41). The PXRD pattern of the same sample indicates that the zirconium oxide layer is amorphous (Supplementary Fig. 42). In contrast, the XPS analysis of the sample heated to 60 °C suggests the absence of zirconium oxide. The combination of these results gives substantial evidence for an amorphous zirconium oxide layer forming on the silica at 100 °C, thus we reasoned this might be why the AuBP-MOF composite only formed at elevated temperatures. To complete the puzzle, we tested whether the AuBPs with the Zr shell could attach to the MOF. The zirconium oxide-coated AuBPs were utilized as the photothermal agents in UIO-66 formation reaction at 100 °C (Supplementary Fig. 43). TEM images showed UIO-66 particles readily attached to the Zr-modified AuBPs, indicating that this may be the mechanism by which the AuBPs are embedded into the UIO-66. Finally, we tested whether the photothermal activation of the AuBPs had an effect on the formation of the Zr oxide shell. AuBPs that were heated to 100 °C as described before but using a hotplate had approximately 50% less zirconium attached to them at all three temperatures (Supplementary Fig. 37).

This unexpected phenomenon enables the insertion of AuBPs into the MOF matrix. Moreover, it is a means to control the number of nanoparticles in the final product by setting the reaction temperature accordingly. As the reaction temperature is set by regulating the exposure to NIR, the amount of incorporated AuBPs could be tuned by simply controlling LED irradiation. Thus, carrying out a reaction at 60 °C enabled separating the suspended AuBPs from the formed MOF and reusing them for any desired purpose. To demonstrate the

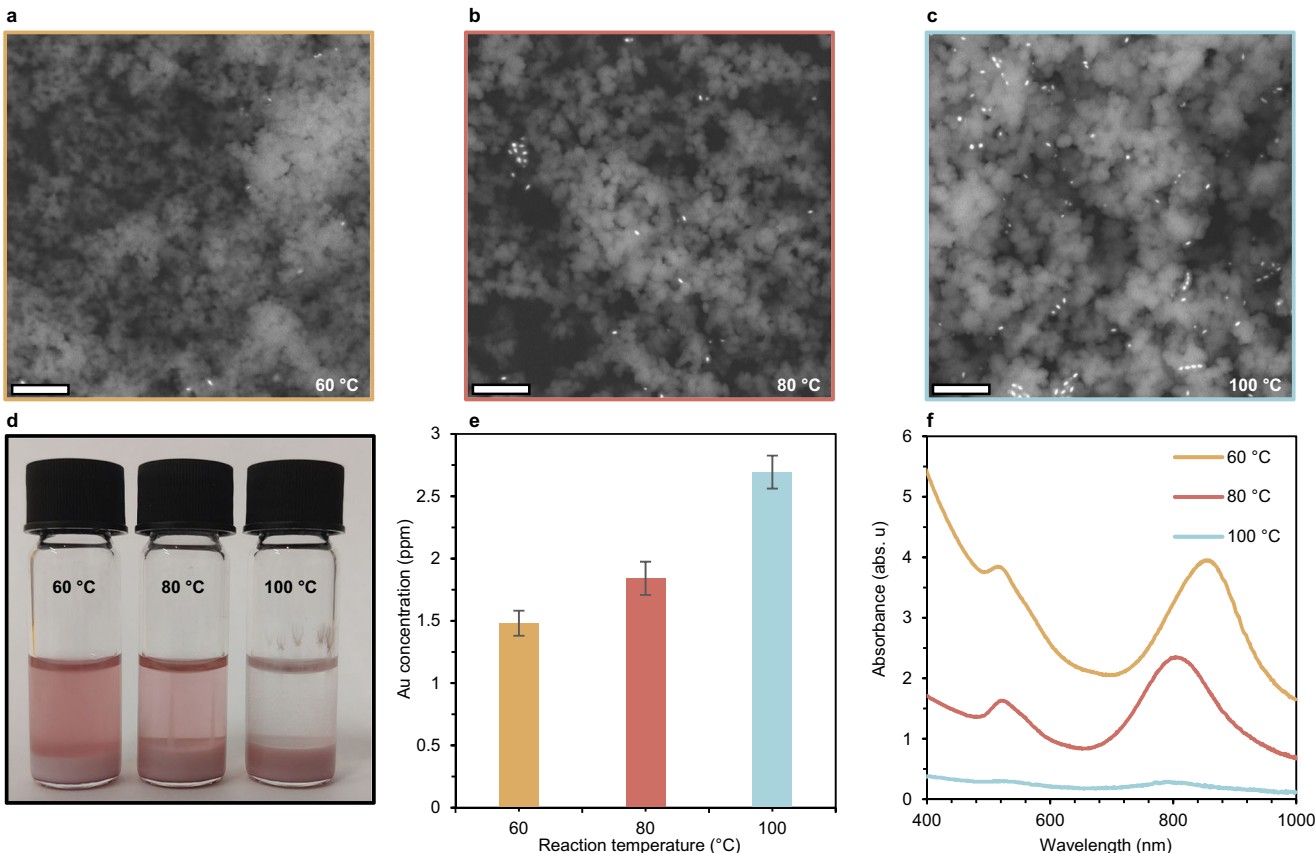

**Fig. 2 | Controlling the insertion of AuBPs into UIO-66. a–c** SEM images of UIO-66 synthesized photothermally at different temperatures (60–100 °C); the embedded AuBPs can be seen as bright dots. The white scale bar size is 1 μm. **d** Images of three reaction vials post-synthesis. **e** ICP-OES results of UIO-66 samples synthesized at different temperatures. Error bars derived from standard deviation according to supplementary table 4. **f** UV–vis spectra of reaction supernatants post-synthesis at different temperatures.

efficiency of the recycling process we performed three cycles of UIO-66 photothermal synthesis at 60 °C, all yielding similar amounts of product (Supplementary Fig. 49). Recycling the AuBPs considerably decreases the energy cost of synthesizing them, improving the sustainability of the photothermal method.

### Versatility and scope of the photothermal synthesis

After establishing a light-induced synthesis for UIO-66, we aimed to test the generality of the photothermal method. Thus, we attempted to synthesize three typical MOFs (r-MIL-88a, HKUST-1, and MOF-5), all having distinct metal ions ($Fe^{3+}$, $Cu^{2+}$, and $Zn^{2+}$) and involving different reaction conditions (see the "Methods" section for details) (Fig. 3a–c). Product characterization via PXRD and SEM validated that all three reactions yielded the expected MOF, demonstrating that the photothermal synthesis is not limited to a specific case but can potentially be applied to any MOF. Importantly, the silica layer encapsulating the AuBPs plays a key role in stabilizing the structure and dispersing the nanoparticles, allowing for a robust and versatile method. Furthermore, to test the possibility of alternative light-to-heat converters, we utilized different photothermal materials (carbon black, graphene/graphene oxide, activated charcoal, $AuBP_{660}$, gold nanospheres, and gold nanorods, see Supplementary Figs. 50–71) to generate the necessary heat to synthesize UIO-66. All reactions were successful, indicating that any inert photothermal agent that can be stabilized under the reaction conditions could be used to produce MOFs. When experimenting with the different types of photothermal materials, we used $AuBP_{660}$, i.e., nano gold bipyramids activated with 660 nm light, to explore the possibility of tuning the wavelength needed for MOF synthesis by simply controlling AuBP size. Similarly, we utilized silica-

encapsulated gold nanospheres (AuNS) and gold nanorods (AuNR). The successful visible light synthesis showed the modularity of the photothermal method where the desired MOF, the identity of the photothermal agent, and its activation wavelength are adjustable. We also utilized carbon-based photothermal materials (carbon black, charcoal, and graphene) with a strong absorbance across the UV–Vis–NIR spectra, allowing activation of the photothermal synthesis at any wavelength without the need to change photothermal materials that require specific wavelengths.

### AuBP-embedded plasmonic MOF

The research of plasmonic MOFs (PMOFs) is a relatively new branch in the broader field of nanoparticle-MOF composites. PMOFs are defined in the literature as composite materials including plasmonic nanoparticles and a MOF, where the hybrid compound retains the unique crystalline MOF structure along with the plasmonic characteristics[46]. There are few synthetic pathways used to fabricate PMOFs. A common route is the one-pot synthesis, where both the MOF and PNPs are synthesized in situ[47]. Multistep approaches also exist, e.g., synthesizing PNPs in an environment where the MOF is present (impregnation)[48], or, in contrast, forming the MOF around the PNPs (seeded growth)[49]. Here, we present a concept, utilizing the plasmonic photothermal ability of PNPs to initiate MOF formation, affording a composite that can be classified as a plasmonic MOF.

Surprisingly, initial experiments testing the photothermal properties of a washed and dried $AuBP_{850}$-embedded UIO-66 ($AuBP_{850}$@UIO-66), where the composite was irradiated with IR light, produced elevated temperatures surpassing 200 °C (Fig. 4a). Encouraged by the massive applicative potential of a photo-responsive

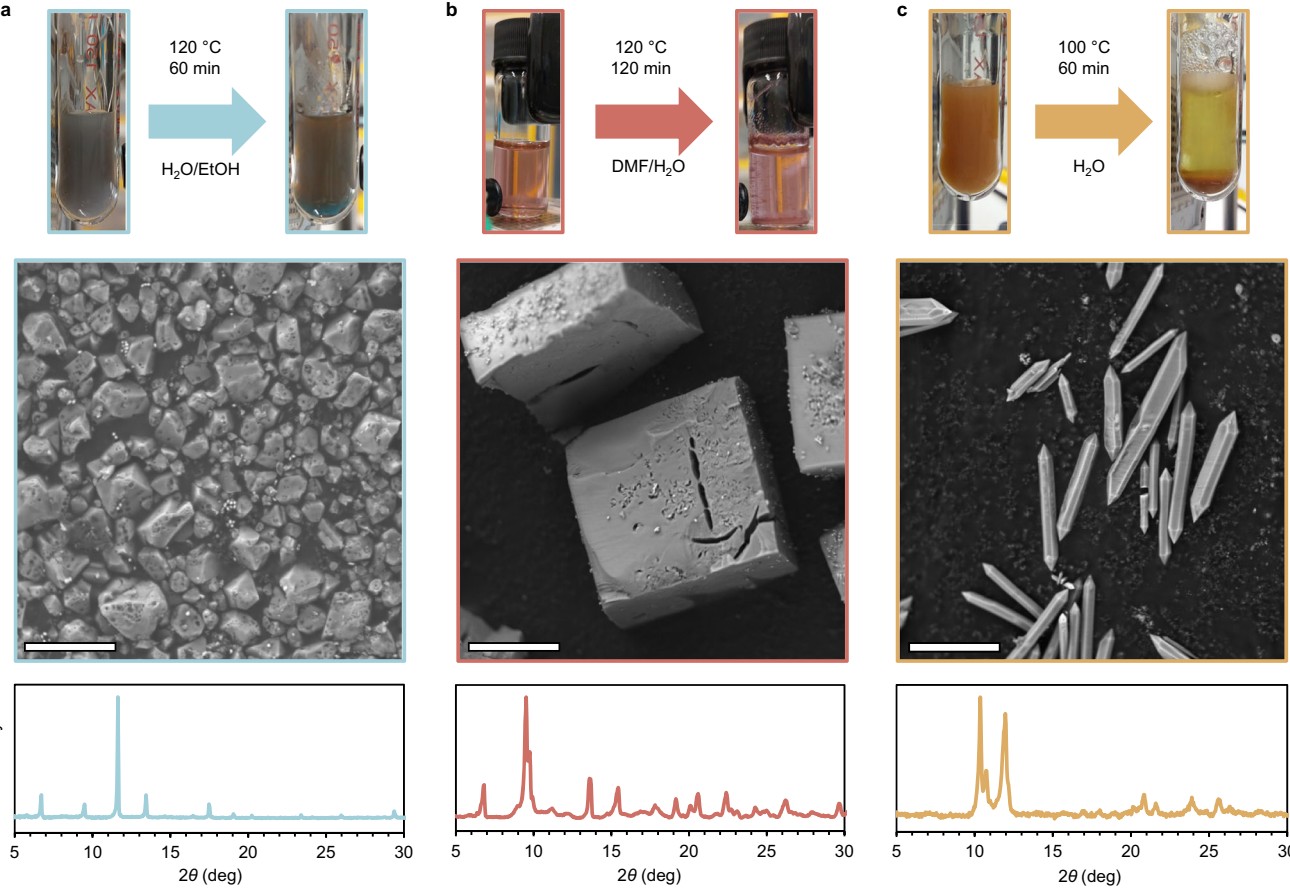

**Fig. 3 | PPR scope. a** Photothermal synthesis scheme, SEM image, and PXRD pattern of HKUST-1[62]. **b** Photothermal synthesis scheme, SEM image, and PXRD pattern of MOF-5[63]. **c** Photothermal synthesis scheme, SEM image, and PXRD pattern of MIL-88A[64]. The white scale bar size is 5 μm.

MOF[50,51], we set out to characterize the photothermal abilities of the composite and test it for different applications. First, we prepared three AuBP$_{850}$@UIO-66 samples, as described above, with varying concentrations of AuBP$_{850}$ (1, 2, and 5 OD in the initial reaction vial) (see Supplementary Movie 1 for the 5 OD synthesis). Subsequently, the samples were washed and dried, and then subjected to irradiation with 850 nm light. The temperature profile during irradiation was recorded and is shown in Fig. 4a. Notably, the plasmonic UIO-66 synthesized with a 5 OD concentration of AuBP$_{850}$ reached nearly 250 °C in under 5 min of exposure. The AuBP$_{850}$@UIO-66 prepared with lower concentrations of nanoparticles also responded to the IR light, reaching well over 100 °C. To evaluate the stability of the photothermal feature over time, we cycled the temperature of a 20 OD-AuBP$_{850}$@UIO-66 between 40 and 180 °C and 40 and 100 °C for three consecutive hours, by turning the NIR LED on and off (Fig. 4c). The results indicate excellent stability throughout both cycling experiments; the first and last cycles were completed in approximately the same times. In addition, very steep heating ramps of 40–180 °C in an average of 22 s, were observed. Furthermore, PXRD and BET analyses of AuBP$_{850}$@UIO-66 were performed after the cycling experiments and ensured that the crystalline MOF structure remained intact (Supplementary Figs. 73, 74), highlighting the immense applicative potential of the plasmonic photothermal MOF.

## Exploring applications of the photothermal MOF

Careful inspection of the thermal profiles presented in Fig. 4a revealed an inflection point at around 80 °C, recurring in all three curves during the heating process. We suspected this might be due to the evaporation of leftover solvents trapped in the MOF matrix. Thus, we

envisioned that a photothermal MOF could be highly useful in the photoactivated release of solvents or target molecules adsorbed by the MOF. Another idea we had from the early stages of the project was to utilize AuBP$_{850}$@UIO-66 as a photothermal agent to induce heat-activated reactions, similar to how we used the AuBPs, but now the inherent catalytic properties of the MOF would be functional.

Utilizing MOFs to produce potable water in arid environments by adsorbing moisture from the air is a promising idea that has shown great applicative potential[52]. Consequently, the light-induced desorption capabilities of the PMOF were tested by adding increasing amounts of water to AuBP$_{850}$@UIO-66, irradiating the PMOF with an 850 nm LED, and analyzing the temperature profiles. The inflection point spotted in the experiments mentioned above re-emerged at roughly the same temperature for all samples, except for PMOF, which was kept dry. Furthermore, increasing the initial volume of water adsorbed by the MOF extended the plateau caused by the evaporation of the adsorbate up to a saturation point, where the amount of water the MOF could hold was exceeded (Fig. 5a).

To compare the photothermal desorption mechanism against desorption using conventional heating, AuBP$_{850}$@UIO-66 and UIO-66 samples were loaded with identical amounts of water and subjected to both heating methods. For photothermal water release, the samples were irradiated for one minute to reach 85 °C (slightly higher than the inflection point observed earlier) using an 850 nm LED, and for conventional heating, the samples were kept in an oven set to 95 °C for the same duration. Notably, AuBP$_{850}$@UIO-66 exposed to NIR released the entire amount of water within a minute. In contrast, the control samples failed to release even a quarter of the adsorbed water, showcasing the impressive photothermal activity of the plasmonic MOF. To

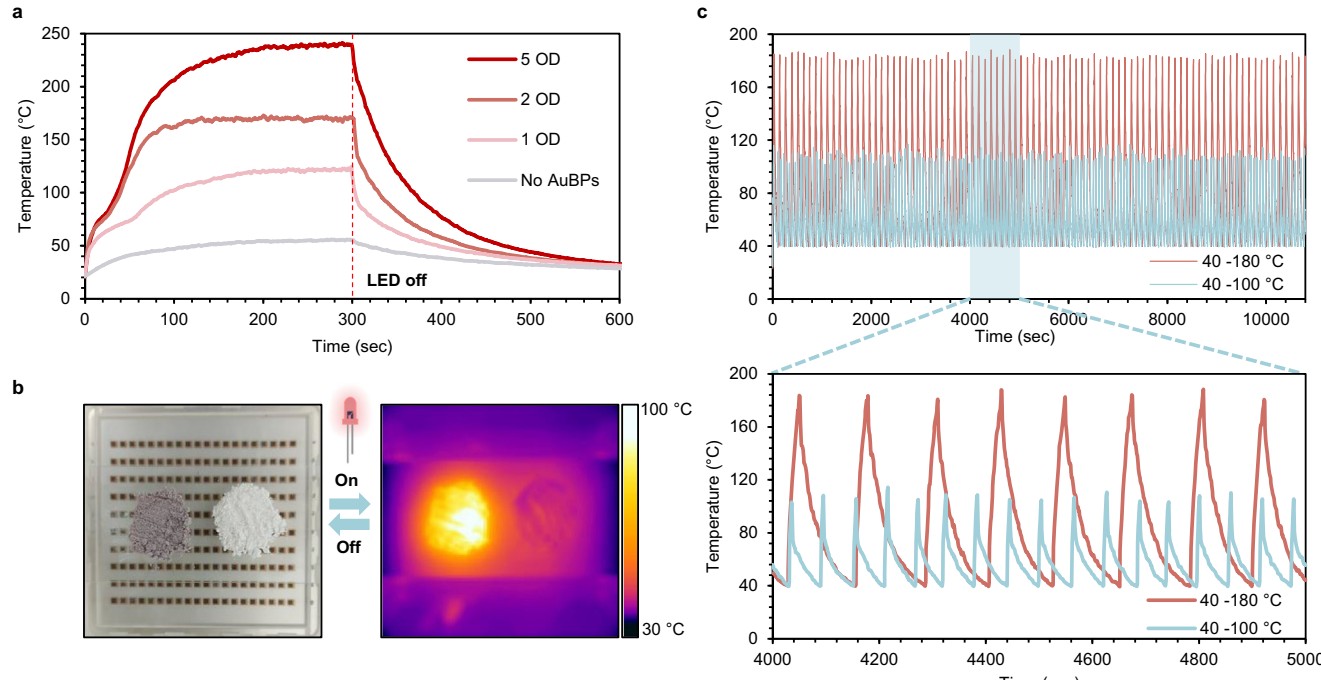

**Fig. 4 | Photothermal converter Plasmonic MOF. a** Temperature profiles of AuBP@UIO-66 with different concentrations of AuBPs irradiated by 850 nm LED. $AuBP_{850}$ concentration is defined via optical density (OD). **b** IR and regular images of AuBP@UIO-66 and UIO-66 before and during IR LED irradiation. LED size-5 cm$^2$ (see Supplementary Movie 2). **c** Heating–cooling cycles of AuBP@UIO-66 (20 OD) between 40 and 180 and 40 and 100 °C.

comprehend the differences between the two desorption techniques, the UIO-66 sample was left in the oven until completely dry, ultimately taking 45 minutes to fully desorb the water (Supplementary Fig. 76). For further examination, we conducted similar experiments using conventional heating at higher temperatures. The results revealed that at 120 °C, the desorption rate became comparable to the photothermal desorption rate observed at 85 °C. The impressive results obtained for the light-induced desorption led us to look for additional ways to exploit the photothermal feature of the composite. Thus, we decided to try "activating" the MOF via the photothermal response. The porous structure of the MOF can adsorb solvents, moisture, and other molecules during synthesis; therefore, it is crucial to "activate" the MOF, i.e., expelling the adsorbate, before using it[53]. Various methods for MOF activation have been reported[54] including a light-induced approach[55]; the standard procedure is to exchange solvents via repeated centrifugation to a more volatile option and simply placing the MOF in a vacuum oven for prolonged periods[56]. To determine whether the proposed method can efficiently expel the solvent, two identical samples of AuBP$_{850}$@UIO-66 (5 OD) were activated in two different ways. First, a conventionally activated sample was put under a vacuum at 120 °C for 17 h[44], whereas the photothermal activation was carried out by exposing the MOF to NIR under a vacuum. To ensure that the activation was complete, the $S_{BET}$ curve was examined (Fig. 5c). Surprisingly, the photoactivation of AuBP$_{850}$@UIO-66 was complete after only 5 min, an enormous improvement over the current conventional procedures.

Finally, the photothermal AuBP$_{850}$@UIO-66 was utilized as a heat source to initiate UIO-66 formation. First, the plasmonic MOF was added to a UIO-66 precursor solution, the solution was irradiated with NIR light, heating the reaction, resulting in UIO-66 synthesis. The product could not be separated from the initial AuBP$_{850}$@UIO-66, essentially affording a photothermal MOF with a lower concentration of AuBPs. This process could be repeated at least four times, significantly increasing the amount of MOF that can be synthesized from

a given amount of AuBPs (Supplementary Table 8). Coupling the MOFs' excellent catalytic properties with the ability to create intense heat at their surface can potentially provide very efficient procedures to be carried out by the presented methodology.

Our current work enabled the development of a light-induced MOF synthesis utilizing photothermal materials. This procedure is robust, versatile, and rapid, making it an ideal alternative to conventional time- and energy-consuming solvothermal syntheses. Furthermore, AuBPs used to generate the heat needed for UIO-66 formation could be embedded into the MOF in situ, through the linking of zirconium to the silica shell, affording AuBP$_{850}$@UIO-66, thus demonstrating a concept for incorporating well-defined nanoparticles into MOF matrices. Perhaps the most exciting feature of the MOF-NP composite is that AuBP's photothermal abilities are retained, yielding a photoresponsive MOF. Importantly, repeated activation of the photothermal response for up to 3 h did not affect the efficiency of the heat generated or the MOF structure, namely, the $S_{BET}$ and PXRD results remained unaltered. The combination of light-to-heat conversion, along with the unique properties of MOFs, may serve as a cornerstone for a wide range of potential applications. To highlight the vast impact of what is possible, photothermal desorption, MOF activation, and catalysis were performed, affording exciting results. Possibly the most impressive result is that MOF activation can be completed within a few minutes, whereas standard procedures can take up to 24 h. Naturally, we expect to continue investigating the photothermal MOF, testing new opportunities for applications, and expanding our insights into the interactions between PNPs and MOFs.

## Methods
### Synthesis of AuBPs
Gold bipyramids were synthesized via seed-mediated growth method described by Sánchez-Iglesias et al.[28]. Silica-coated AuBPs were synthesized according to the procedure described by Lemcoff et al.[29].

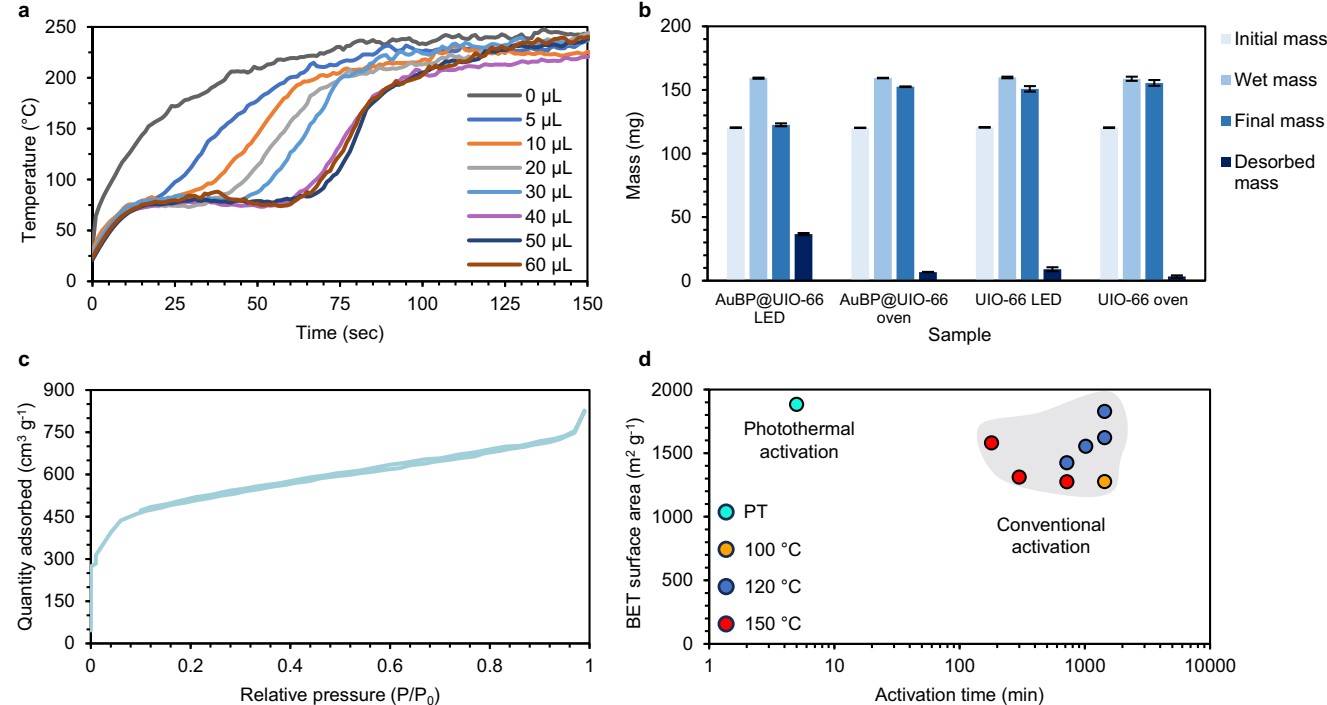

**Fig. 5 | Photothermal activation. a** Temperature profile of 22 mg of AuBP@UIO-66 that was treated with different volumes of water and irradiated by 850 nm LED. The volume of the water is measured with microliters (μL). **b** Recorded masses of AuBP@UIO-66 and UIO-66 before and after drying, using photothermal and conventional heating. Error bars represent the standard deviation according to the three samples that were tested. **c** $N_2$ adsorption isotherm of AuBP@UIO-66 activation via the photothermal method. **d** A surface area comparison between photothermal activation and other reported conventional activation processes (taken from refs. 35,65–70).

### Synthesis of AuNS

Gold nanospheres were synthesized based on a method published by Karakocak et al.[57]. Initially, 10 mg of HAuCl$_4$ was dissolved in 90 mL of TDW and heated to 100 °C. Upon boiling, 400 μL of 250 mM sodium citrate solution was added while vigorously stirring the solution. The solution was aged for 30 min until it turned to a red-wine color. Subsequently, the solution was centrifuged 3 times at 10,000×*g* for 15 min with CTAB 1 mM. The AuNS were encapsulated using the same procedure as the AuBPs.

### Synthesis of AuNR

Gold nanorods were synthesized based on a method published by Jana[58], with slight modification. 1.6 mL of HAuCl$_4$ (6 mM) was added to 8 mL of aqueous solution of CTAB 200 mM. Later AgNO$_3$ (10 mM) and 120 μL of ascorbic acid (100 mM) were added to the solution. Finally, 30–40 μL of NaBH$_4$ (1 mM) was added. The color of the solution gradually changed from colorless to a deep brown within minutes. Subsequently, the gold nanorods underwent purification through centrifugation at 15,000×*g* for 10 min, followed by washing with an aqueous CTAB solution (1 mM). Finally, the nanorods were encapsulated using the same procedure as the AuBPs.

### Photothermal synthesis of UIO-66

AuBP@UiO-66 was synthesized via a photothermal reaction based on the reported solvothermal synthesis[35]. Initially, terephthalic acid (123 mg) was dissolved in 10 mL of DMF. Next, ZrCl$_4$ (123 mg) was dissolved in a 10 mL HCl:DMF mixture (1:5) and aged at moderate temperature for 30 min to enhance its dissolving. Finally, the two solutions were mixed into a vial, and varying concentrations of AuBPs were added. During the synthesis, the solution was irradiated with an LED (850 nm, 100 W). The LED operation and temperature monitoring were controlled using LabView software. Reactions involving different photothermal reagents were conducted similarly, with the choice of LED dependent on the reagent's optical absorption. The product was centrifuged (3000×*g*, 5 min) and washed three times with DMF and three times with ethanol. The precipitate was dried in a vacuum oven at 120 °C for 17 h, resulting in dried activated UIO-66.

### Photothermal synthesis of r-MIL-88A

Light-induced synthesis of r-MIL-88A was based on the solvothermal procedure for r-MIL-88A reported by Wang et al.[59]. Initially, 0.4 mmol of FeCl$_3$·H$_2$O and 0.4 mmol of fumaric acid were dissolved individually in 1 mL of ultra-pure water. Then, the two solutions were mixed, and AuBPs were added. The solution was then irradiated with a 100 W 850 nm LED for 60 min at 100 °C.

### Photothermal synthesis of HKUST-1

The light-induced synthesis of HKUST-1 was performed based on a microwave-assisted synthesis of HKUST-1 reported by Seo et al.[60]. First, 0.083 mmol of H$_3$BTC and 0.152 mmol of Cu(NO$_3$)$_2$·3H$_2$O were dissolved in 1 mL of a H$_2$O:EtOH 1:1 mixture. The two solutions were mixed, and AuBPs were added. The resulting solution was then transferred to a pressure tube obtained from Ace Glass. Finally, the solution was irradiated with a 100 W 850 nm LED for 60 min at 120 °C.

### Photothermal synthesis of MOF-5

The light-induced synthesis of MOF-5 was based on a procedure reported by Chen et al.[61]. Briefly, 0.148 mmol of Zn(NO$_3$)$_2$·6H$_2$O and 0.018 mmol of H$_2$BDC were each dissolved in 1 mL of DMF. Then, the two solutions were mixed, and 18 μL of ultra-pure water and AuBPs were added. Finally, the solution was irradiated with a 100 W 850 nm LED for 120 min at 120 °C.

### Photothermal activation of UIO-66

Dried powder from AuBP@UIO-66 was placed into a tube, and the tube was sealed and connected to an air pump. Then, the powder was

irradiated with a 100 W 850 nm LED; simultaneously, the air pump was operated for 10 min.

### UiO-66@UiO-66 synthesis process

Dried AuBP@UIO-66 powder was dissolved in UIO-66 precursor solution; then the solution was irradiated with a 100 W 850 nm LED for 30 min at 100 °C. The product was washed, dried, and then dissolved in a new UIO-66 precursor solution to begin a new cycle as described. This was repeated four times.

### Reporting summary

Further information on research design is available in the Nature Portfolio Reporting Summary linked to this article.

## Data availability

All the data can be found in the paper and the supplementary information. Source data are provided with this paper.

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

## Acknowledgements

We thank Dr. Einat Nativ-Roth for taking the SEM images, and Dr. Vladimir Ezersky for assistance with JEM-2100F TEM. Y.W. acknowledges the support of the Zuckerman STEM Leadership Program, and the Israel Science Foundation (ISF), grant No. 2491/20. O.S. acknowledges funding from the School of Sustainability and Climate Change scholarship.

## Author contributions

YW and IH supervised the project. OS conducted all the experiments. EY, NL, and OS set up the LED temperature regulation system. NL, DY, AB, and OS synthesized the nanoparticles. OS took the SEM images. AB took TEM images. OS and RS conducted the BET analysis. RS performed the PXRD analysis. NL, OS, and YW wrote the manuscript.

## Competing interests
The authors declare no competing interests.
