## [Peer Review File · Nature Communications]

Reviewers' comments:

Reviewer #1 (Remarks to the Author):

The paper by Shelonchik et al. deals with the light induced MOF synthesis. After reading the manuscript, although the results may be interesting I found a lack of proper characterization of the composite materials. In addition, the selected plasmonic material need to be fully addressed. In addition, the photothermal synthesis have been demonstrated with any absorbing media. In my opinion, the scientific impact is somehow limited. In what follows, a number of major points to support my recommendation is listed;

- It is known that Au bypyramids present narrow optical responses but it is also know that the extinction spectra of gold nanoparticles is the sum of the contribution of the absorption and the scattering part of the spectrum. Therefore, if the photothermal effect of Au nanorods requires higher absorption spectrum, why not use smaller nanords?. The photothermal efficiency need to be demonstrated varying the absorption and scattering part of the extinction spectrum.

- In the manuscript, it is written that the Au BPs are coated with silica to enhance their stability but, how the silica shell thickness affects the efficiency of the photothermal effect?, especially considering that the silica layer may act insulator. From the TEM shown in the supplementary material the silica shell is close to 30-40 nm, which clearly means that there should be room for improvement in the photothermal conversion.

- In general, I found a lack of proper characterization of the composite materials. For instance, in line 118-119 it is written that the AuBPs may precipitate with the product or remained dispersed in the supernatant, later in line 125-126 it is hypothesized that larger MOFs could entrap the AuBPs but this is nor demonstrated in the manuscript by either the SEM of the TEM characterization. Why is this entrapment?, are the MOFs coating the particles?, if so, why?. This section is not convincing. From the SEM characterization is not easy to check if the particles are overcoated by the MOF

On the other hand, it is also written that under certain conditions the BPs remain in the supernatant, and again there is no characterization of the plasmonic nanoparticles after the MOF synthesis under any conditions.

- I do not agree with the concept of scale up, if the volume is only 20 mL!!, which is the increase in volume from the initial experiments to the "scale up".

- The characterization of the optical properties of the composites shows a red-shift in the position of the LSPR with the temperature, how is this possible?. This shift cannot be assigned to a change in the refractive index since the AuBPs are coated with a thick silica shell that should provide a homogeneous refractive index with no matter of solvent or further coating. Again a proper characterization may help.

- The photothermal conversion of AuBPs should be also compared with other shaped nanoparticles. For instance, spherical Au NPs with an average diameter of 60 nm, also coated with silica and OD of around 2.

Reviewer #2 (Remarks to the Author):

The manuscript "Light-induced MOF synthesis enabling composite photothermal materials" reports a new strategy for MOF synthesis utilizing LSPR effect of plasmonic materials to generate hotspot for acceleration of MOF formation. The manuscript also describes a versatile of MOFs and plasmonic materials that can be employed for this strategy, and the authors propose several potential applications for the synthesized plasmonic MOF materials.

The photothermal synthesis demonstrated in the manuscript is advantageous over conventional heating method because of its energy efficient and upscale potential. The good versatility of this strategy is beneficial for making other plasmonic MOF materials for different purposes. However, I recommend conducting additional experiments and including additional supporting data to further confirm the mechanism of light-induced MOF synthesis. The manuscript is not suitable for publication at the current stage.

My other comments:

1. In page 2, line 79, it is claimed that "AuBPs were encapsulated to prevent deterioration of the bipyramidal structure, causing a blue shift of the wavelength". Please provide experimental data or relevant reference for this observation.
2. In figure 1b-c, it appears that AuBPs solutions of different OD reach consistent temperature after 5 minutes, however there is a different lag time for the reaction to happen (0 minute for 2 OD at 120oC and 10 minutes for 0.5 OD at 90oC). Please explain. In addition, the reaction rate seems to be the same at different temperature (the slopes before reactions reach maximum yield are similar). Please explain.
3. Do you have the yield profile for conventional heating? Does the reaction only reach the maximum yield overnight?
4. To better confirm the hypothesis that under light irradiation, AuBPs generate hotspot that accelerate the MOF formation, it is recommended that an additional experiment should be conducted by using a compound/materials capable of absorbing light (preferably at 850 nm) but does not generate hotspot. This is to confirm that the hotspots that are generated cause a locally hot environment for the reaction to take place.
5. In part 2 (Embedding AuBPs in MOF via photothermal synthesis), it is claimed that larger MOF particles can entrap the AuBPs and finer particles are less likely to cover the NPs. Do you have TEM images to confirm that the AuBPs are trapped inside the larger MOFs (at high temperature) and are not trapped inside the finer MOFs (at lower temperature).
6. How does the size of MOFs compare to AuBPs? Please provide a size distribution profile of MOFs formed at different temperature in complement with the SEM images to facilitate the size comparison. In the case of 60oC, if the MOF size is larger than the AuBPs, can you quantify the amount of AuBPs being trapped inside the MOF?
7. In part 3 (Versatility and scope of the photothermal synthesis), please include the reference on PXRD and SEM of the synthesized MOFs to support the experimental PXRD and SEM obtained.

8. The formation of HKUST-1 in part 3 took place in EtOH/H₂O 1:1 solvent. How does the authors reach temperature of 120°C while the solvent boiling point is around 80-90°C?

9. Similarly, in part 4 (AuBP-embedded plasmonic MOF), line 182, the plasmonic UiO-66 reached 250°C, although DMF boiling point is 153°C. Please explain.

10. In part 5 (Exploring applications of the photothermal MOF), the mechanism for acceleration of water/solvent desorption remains unclear. Both methods rely on heat-induced evaporation of water/solvent, hence the rate of desorption should be limited to the evaporation of water/solvent molecules. If the higher temperature at the hotspot cause the acceleration of evaporation/desorption, it should also be possible to achieve similar rate of desorption by increasing the temperature of the conventional heating process. Please elaborate on how the generation of hotspot can accelerate the water/solvent desorption.

Reviewer #3 (Remarks to the Author):

in this manuscript Weizmann et al. report the light induced MOF synthesis due to the photothermal effect of gold nanoparticles and the resulting temperature increase on the reaction media. the resulting composite show also photothermal effect and this effect can be applied for the fast activation of MOFs. first i would like to highlight that some important ref. on the photothermal effect on MOFs are missing. especially the one from espin et al. (10.1021/acsami.8b00557) is important because the photothermal effect (using UV light) was used for the fast activation of different MOFs such as HKUST-1, Zif-67, UiO-66-X, etc... the same advantage that authors claim in this manuscript. there are also others...

the in situ synthesis of MOFs thank to the heat generated by gold nanoparticles is interesting but i would like to know how the resulting composite is... how the gold is distributed in the MOF. Microscopy is not very clear about this.

the other question that i have is about the need of this kind of synthesis to have this fast activation, did the authors try to mix the nanoparticles with already made MOF particles and activate them?

did the authors try to synthesis core Shell particles with this approach?

to conclude i have my doubts about the novelty (specially considering this previous study on fast photothermal activation) and i would like to see a reviewed manuscript before to consider the acceptance of this manuscript in nature communications

We extend our sincere gratitude to the reviewers for their invaluable and insightful comments. This feedback has significantly enriched our manuscript, allowing us to incorporate essential data and present a more comprehensive research study. We appreciate the opportunity to address the points raised in the comments and we hope that we were able to clarify any misunderstandings or gaps in our work.

Reviewers' comments:

Reviewer #1 (Remarks to the Author):

The paper by Shelonchik et al. deals with the light induced MOF synthesis. After reading the manuscript, although the results may be interesting I found a lack of proper characterization of the composite materials. In addition, the selected plasmonic material need to be fully addressed. In addition, the photothermal synthesis have been demonstrated with any absorbing media. In my opinion, the scientific impact is somehow limited. In what follows, a number of major points to support my recommendation is listed;

Response 1- general summary: We would like to express our sincere appreciation to reviewer #1 for taking interest in our findings and dedicating time to reviewing our manuscript. The valuable insights and comments provided by the reviewer significantly contributed to the completion of this study. Firstly, we acknowledge the lack of comprehensive characterization of the composite materials, as suggested by the reviewers, we have now conducted further investigations to elucidate the mechanism by which the hybrid material is formed. This crucial aspect was missing in our initial manuscript, and we are genuinely grateful for bringing it to our attention.

As elaborated in subsequent sections, our decision to primarily focus on AuBP₈₅₀ originates from their high heat conversion efficiency and ease of tuning their size, as well as controlling the LSPR peak. However, we have demonstrated in the manuscript that our method is adaptable to other plasmonic particles as well. This versatility makes our approach more universally applicable and user-friendly. We firmly believe that the scientific community at large, particularly the MOF community, maintains a significant interest in alternative MOF syntheses. Consequently, we believe that our photothermal synthesis holds the potential for creating a substantial impact and even finding extensive usage following its publication.

We thank the reviewer once again for the valuable feedback and for contributing to the refinement of our manuscript. We hope that our revised work provides the clarity and depth that was previously missing, further highlighting the importance of our paper.

- It is known that Au bypyramids present narrow optical responses but it is also know that the extinction spectra of gold nanoparticles is the sum of the contribution of the absorption and the scattering part of the spectrum. Therefore, if the photothermal effect of Au nanorods requires higher absorption spectrum, why not use smaller nanords?. The photothermal efficiency need to be demonstrated varying the absorption and scattering part of the extinction spectrum.

Response 1-1: Firstly, it is known that anisotropic elongated nanoparticles, such as bipyramids and rods, exhibit excellent photothermal conversion compared to other gold nanostructures. Thus, both geometries are suitable choices for our photothermal method. Additionally, there is evidence from research, such as the study conducted by Zhang *et al.*¹, suggesting that bipyramids demonstrate superior heat conversion capabilities compared to nanorods.

Indeed, smaller anisotropic gold nanoparticles tend to exhibit reduced scattering,² making them potentially more advantageous for photothermal applications. In our research, we have demonstrated the utilization of smaller nanoparticles (AuBP₆₆₀) in the photothermal synthesis of MOFs, as shown in Supplementary Fig. 59. Based on the literature, these smaller particles can offer enhanced photothermal efficiency.

However, it is important to highlight that our method is not limited to a specific nanoparticle size or shape. In our manuscript, we have showcased the successful synthesis of MOFs using two sizes of AuBPs and carbon-based materials, and thanks to reviewer #1 comments, we even used gold nanorods and gold nanospheres as shown later in response 1-6. This versatility underscores the wide range of particles that can be employed based on specific research objectives. Furthermore, as described in the manuscript, we have generally constrained the maximum temperature during the synthesis process, resulting in the underutilization of the full photothermal conversion potential of the nanoparticles. Therefore, the potential improvement in heat conversion becomes less critical for us from this perspective.

The main reason for choosing AuBPs as our primary particle of interest lies in our well-established protocol, which allows us to produce large quantities of monodisperse AuBPs with precise size control. This advantage enables consistency and scalability in our experiments.

As previously mentioned, we opted to synthesize and encapsulate gold nanorods (AuNRs) that exhibit absorption at 730 nm. We utilized these AuNRs to conduct a photothermal synthesis of UiO-66, and the successful outcomes are shown in supplementary section 13.6.

We appreciate the valuable insights provided by reviewer #1, and we hope this response clarifies the various aspects of our research related to nanoparticle size, shape, composition, photothermal efficiency, and the flexibility of our method.

- In the manuscript, it is written that the Au BPs are coated with silica to enhance their stability but, how the silica shell thickness affects the efficiency of the photothermal effect?, especially considering that the silica layer may act insulator. From the TEM shown in the supplementary material the silica shell is close to 30-40 nm, which clearly means that there should be room for improvement in the photothermal conversion.

Response 1-2: The thick silica shell plays a crucial role in maintaining the stability of the AuBPs. Given the harsh conditions under which the reaction occurs, including a very acidic environment and high temperatures, the AuBPs would dissolve during the processes. Consequently, without a substantial silica shell, heat conversion be unattainable, and thus heat dissipation would also be compromised. Additionally, the silica shell serves a dual purpose by enhancing dispersibility in the solution and thwarting aggregation, which in turn further facilitates efficient heat dissipation.

Furthermore, it is noteworthy that the presence of a mesoporous silica shell seems to have no detrimental effect on the photothermal efficiency of plasmonic nanoparticles. In fact, there are studies, such as the work by Yang *et al.*³, which suggest that the addition of a mesoporous silica shell can enhance the heat dissipation from the plasmonic nanoparticles, even with shell thickness in the tens of nanometers range.

Thus, from our analysis of literature reports, it appears that the presence of a mesoporous silica shell does not act as an insulator that hampers the photothermal conversion of AuBPs. Instead, it can contribute positively to the overall photothermal performance of the nanoparticles.

- In general, I found a lack of proper characterization of the composite materials. For instance, in line 118-119 it is written that the AuBPs may precipitate with the product or remained dispersed in the supernatant, later in line 125-126 it is hypothesized that larger MOFs could entrap the AuBPs but this is not demonstrated in the manuscript by either the SEM or the TEM characterization. Why is this entrapment?, are the MOFs coating the particles?, if so, why?. This section is not convincing. From the SEM characterization is not easy to check if the particles are overcoated by the MOF. On the other hand, it is also written that under certain conditions the BPs remain in the supernatant, and again there is no characterization of the plasmonic nanoparticles after the MOF synthesis under any conditions.

Response 1-3: We want to thank reviewer #1 for this essential comment, as well as reviewers #2 and #3, who raised concerns about the characterization and the formation mechanism of the AuBP@UIO-66 composite. The reviewers' insightful comments prompted us to conduct further investigations and thus recognize the gaps in this particular aspect of our study. As a result, we have made significant progress in improving our understanding of the composite's formation and structure.

As mentioned in our pre-revised manuscript, initially, our hypothesis focused on the size of the MOF particles and how larger particles may entrap AuBPs. However, through subsequent experiments (as explained in detail in response 2-6), we have realized the limitations of this idea. The size of the MOF particles appears to have a less significant impact on the composite formation than we initially thought.

We have now added to our study several experiments aimed to better characterize the composite's structure and elucidate the temperature sensitive formation mechanism. First, further characterization of the composite, synthesized at different temperatures, can now be found in the form of TEM images (Supplementary section 7.5). The TEM images clearly show that when the MOF formation reaction is carried out at 100 °C the AuBPs attach to the UIO-66 particles creating the hybrid material. In contrast, lower reaction temperatures lead to AuBPs being separated from the MOF.

Second, we have now conducted several experiments to understand the interaction between the AuBPs and the MOF. We started by probing whether MOF precursors (ZrCl₄ or BDC) could bind to the silica surface at elevated temperatures, enabling the AuBP-MOF interaction. We did this by preparing solutions identical to those used for photothermal synthesis of UIO-66 excluding one of the precursors. These solutions were heated to different temperatures (60, 80, 100 °C) and the AuBPs were washed and isolated for further characterization. Interestingly, the AuBP solution mixed with ZrCl₄ and

heated to 100 °C exhibited a turbid appearance (Revision Fig. 1b), indicating reduced solubility of the AuBPs in DMF. The samples isolated from solutions where BDC was present were subjected to zeta potential analysis and showed no significant effect to the silica's surface properties. In the case of zirconium chloride, the samples were analyzed by ICP-OES, EDS elemental mapping, STEM imaging and XPS (Revision Fig. 1 and supplementary section 9). The results revealed that at elevated temperatures an amorphous layer of zirconium oxide forms around the AuBPs with the layer shrinking as the temperature drops. Furthermore, we conducted an additional experiment where AuBPs pretreated with $ZrCl_4$ to form the oxide shell, were used to photothermally synthesize UIO-66. TEM images of the product demonstrated that the modified AuBPs were attached to MOF particles (Revision fig. 4). Based on these observations we reasoned that when carrying out the photothermal MOF synthesis at elevated temperatures, the zirconium precursor reacts with the silica surface to afford the Zr oxide layer. This shell can then serve as a link between the MOF and AuBPs enabling the formation of the hybrid as a factor of the temperature. We have now added a thorough discussion of these results to the manuscript.

We sincerely appreciate the valuable feedback from reviewer #1, as well as reviewers #2 and #3, which has driven us to further investigate and improve our understanding of the AuBPs@UIO-66 composite.

Revision Figure 1| Interaction between zirconium to AuBPs' silica shell. a, TEM image of AuBPs@UIO-66 synthesized photothermally at 100 °C for 20 minutes. b, Image of the vials contain the solutions of AuBPs (2 OD) with ZrCl₄ in DMF and HCl, after heating photothermally to different temperatures for one hour. c, ICP-OES results of b. d-f, STEM image and zirconium K series mapping via EDS of b (from left: 60 °C, 80 °C, 100 °C). (White scale bar- 100 nm)

- I do not agree with the concept of scale up, if the volume is only 20 mL!!, which is the increase in volume from the initial experiments to the “scale up”.

Response 1-4: We understand the concern raised about using the term "scale up" in this particular context. To address this concern, we have revised our terminology to refer to the "increased volume" instead.

It is worth mentioning that we are planning to improve our experimental setup to enable unambiguous scaled-up syntheses with larger volumes.

We thank the reviewer for bringing this to our attention.

- The characterization of the optical properties of the composites shows a red-shift in the position of the LSPR with the temperature, how is this possible?. This shift cannot be assigned to a change in the refractive index since the AuBPs are coated with a thick silica

shell that should provide a homogeneous refractive index with no matter of solvent or further coating. Again a proper characterization may help.

Response 1-5: In Figure 2f of the manuscript, we illustrate the observed blue-shifting of the LSPR peaks of the composites synthesized at different temperatures relative to the original LSPR peak of the AuBPs. While it is true that the silica shell covering the AuBPs prevents changes in the refractive index from affecting the LSPR peak, we would like to highlight that the shifting observed is attributed to the harsh synthesis conditions.

During the synthesis process, the solution is highly acidic and requires elevated temperatures. Under these conditions, the tips of the AuBPs undergo slight degradation, even with the presence of the thick silica shell. This degradation leads to a blue shift in the LSPR peak. The UV-vis spectrum further demonstrates that as the temperature increases, the peak undergoes more pronounced shifting.

To provide additional clarity, we have included more TEM images in the supplementary information that specifically focus on the degraded tips of the AuBPs (Supplementary section 7.5).

- The photothermal conversion of AuBPs should be also compared with other shaped nanoparticles. For instance, spherical Au NPs with an average diameter of 60 nm, also coated with silica and OD of around 2.

Response 1-6: To address this point, we synthesized gold nanospheres (AuNS) with an average diameter of 35 nm and coated them with a 40 nm silica shell (see Revision Fig. 2b). The LSPR peak of the AuNS is around 530 nm. To compare the heat conversion, we irradiated 1 ml of the 2 OD AuNS solution in DMF with an 8W LED at a wavelength of 525 nm for 10 minutes. Similarly, we performed the same irradiation on a 1 ml solution of 2 OD AuBPs₈₅₀ in DMF, using an 8W LED at a wavelength of 850 nm. As shown in Revision Fig. 2c, the temperature of the AuBP solution reached around 100 °C, while the AuNS solution only reached a maximum of 60 °C.

To further strengthen our findings, we have included a reference that compares the plasmonic heating abilities of AuBPs to AuNS.¹ These findings support our conclusion that AuBPs exhibit superior photothermal conversion abilities.

In addition to the above, we also attempted to synthesize UIO-66 photothermally using the encapsulated AuNS. We introduced 2 OD of AuNS into the UIO-66 precursor solution and irradiated it with a 525 nm LED at 100 W, maintaining a temperature of 100 °C. Successfully, UIO-66 was synthesized as confirmed by the PXRD spectrum (see Revision Fig. 2d and Supplementary section 13.5).

We are grateful to reviewer #1 for encouraging us to synthesize and encapsulate nanospheres, as this led us to expand our scope and investigate the photothermal synthesis of MOF with them.

Revision Figure 2| AuNS heating conversion. a, UV-vis spectrum of AuNS. b, TEM image of AuNS. c, Temperature profiles of AuNS and AuBPs₈₅₀. 1 ml solution of 2 OD AuNS was irradiated with an 8W LED at 520 nm. 1 ml solution of 2 OD AuBPs was irradiated with an 8W LED at 850 nm. d, PXRD spectrum of AuNS@UIO-66.

Reviewer #2 (Remarks to the Author):

The manuscript “Light-induced MOF synthesis enabling composite photothermal materials” reports a new strategy for MOF synthesis utilizing LSPR effect of plasmonic materials to generate hotspot for acceleration of MOF formation. The manuscript also describes a versatile of MOFs and plasmonic materials that can be employed for this strategy, and the authors propose several potential applications for the synthesized plasmonic MOF materials.

The photothermal synthesis demonstrated in the manuscript is advantageous over conventional heating method because of its energy efficient and upscale potential. The good versatility of this strategy is beneficial for making other plasmonic MOF materials for different purposes. However, I recommend conducting additional experiments and including additional supporting data to further confirm the mechanism of light-induced MOF synthesis. The manuscript is not suitable for publication at the current stage.

Response 2- general summary: *We sincerely appreciate reviewer #2 for dedicating time and effort to review our manuscript. These comments have been valuable in guiding us towards a more complete manuscript. The suggestion to conduct additional experiments*

and include supporting data to further confirm the mechanism of light-induced MOF synthesis was particularly insightful. In response to this suggestion, we conducted the recommended experiments, which provided us with a deeper understanding of the mechanisms governing the formation of the detailed composite. The revised manuscript now includes these findings, aligning with the reviewer's valuable input.

We are grateful for reviewer #2's contribution in improving the quality of our research and manuscript. It is our hope that the revised manuscript, which addresses the concerns and suggestions, is now suitable for publication.

My other comments:

1. In page 2, line 79, it is claimed that “AuBPs were encapsulated to prevent deterioration of the bipyramidal structure, causing a blue shift of the wavelength”. Please provide experimental data or relevant reference for this observation.

Response 2-1: We appreciate reviewer #2 for drawing our attention to the missing data regarding the encapsulation of AuBPs. In our revised manuscript, we have included references (37,38) that demonstrate the significance of the silica shell in preserving the structural integrity of gold nanoparticles. In addition, we conducted an experiment in which uncoated AuBPs were heated, and TEM images were taken before and after the procedure. As shown in Revision Fig. 3b significant degradation of the bipyramidal structure occurs after heating. Furthermore, the UV-vis spectrum of the particles before and after heating (see revision Fig. 3c) clearly demonstrates the complete disappearance of the LSPR peak after the heating process. These findings provide direct evidence supporting the need for encapsulating our AuBPs.

Revision Figure 3| Uncoated AuBPs. **a**, TEM image of uncoated AuBPs pre-heating. **b**, TEM image of uncoated AuBPs after heating for one hour (White scale bar 20 nm). **c**, UV-vis spectra of uncoated AuBPs before and after heating.

2. In figure 1b-c, it appears that AuBPs solutions of different OD reach consistent temperature after 5 minutes, however there is a different lag time for the reaction to happen (0 minute for 2 OD at 120oC and 10 minutes for 0.5 OD at 90oC). Please explain. In addition, the reaction rate seems to be the same at different temperature (the slopes before reactions reach maximum yield are similar). Please explain.

Response 2-2: Upon careful examination of Figures 1b-c in the manuscript and considering the feedback provided in this comment, we acknowledge the issue that arises from the current presentation. We apologize for the confusion that may have been caused. To address this concern, we have revised Figure 1c by increasing its resolution and adding more measurement points at shorter time intervals. The revised graph now more accurately represents the lag time and reaction slopes for each sample, providing a clearer depiction of the data.

The reason for the different lag times observed among the curves is attributed to the variation in heating rate caused by the different concentrations of AuBPs in each sample. For instance, after 2.5 minutes of heating, the reaction with a concentration of 2 OD reaches an approximate temperature of 130 °C, while the reaction with a concentration of 1 OD reaches about 100 °C. The higher temperature attained in the 2 OD sample initiates the MOF formation process earlier, resulting in a shorter lag time.

We appreciate reviewer #2 for highlighting this discrepancy and guiding us to improve the presentation of the data. The revised graph now accurately represents the experimental

observations, providing a clearer understanding of the lag time and reaction slopes for each sample.

3. Do you have the yield profile for conventional heating? Does the reaction only reach the maximum yield overnight?

Response 2-3: In response to reviewer #2's question, we now included in our supplementary information a yield profile for UIO-66 using conventional heating at different temperatures (Supplementary Fig. 47). As seen from the results, we observed that the reactions reached their maximum yield faster than expected, contrary to the conventional UIO-66 protocols where it typically recommend an overnight incubation.⁴ However, it is important to note that there still exists a substantial time difference between the conventional and photothermal reactions, especially at lower temperatures. For example, at 80 °C, the conventional heating method required 80 minutes to complete the synthesis, whereas the photothermal method achieved the same result in just 30 minutes. This time gap becomes more significant as the temperature decreases. At 60 °C, the conventional heating method took nearly 4 hours, whereas the photothermal method completed the synthesis in just 90 minutes.

These results highlight the efficiency and time-saving benefits of the photothermal method compared to conventional heating. We appreciate this comment for prompting us to conduct the additional experiment and provide a clearer understanding of the reaction profiles.

4. To better confirm the hypothesis that under light irradiation, AuBPs generate hotspot that accelerate the MOF formation, it is recommended that an additional experiment should be conducted by using a compound/materials capable of absorbing light (preferably at 850 nm) but does not generate hotspot. This is to confirm that the hotspots that are generated cause a locally hot environment for the reaction to take place.

Response 2-4: We agree with reviewer # 2 that this is a valuable control experiment. Thus, we carried out a reaction using Methylene blue as a photothermal agent that does not generate hotspots, as the reviewer suggested. Methylene blue has an absorption peak at 660 nm, which is comparable to the AuBP₆₆₀ that were utilized in our work. The experiment was carried out at 60 °C, as the methylene blue limited us from reaching higher temperatures, due to the lower photothermal efficiency of organic dyes.^{5,6} The synthesis took approximately four hours, which is comparable to the reaction time observed with conventional heating (see comment 2.3 and supplementary sections 10 and 11), and significantly longer than the photothermal MOF formation. This result provides further evidence for the importance of the hotspots generated by AuBPs in enhancing the efficiency of MOF synthesis. We are grateful to reviewer #2 for this insightful suggestion, which allowed us to reinforce our hypothesis.

5. In part 2 (Embedding AuBPs in MOF via photothermal synthesis), it is claimed that larger MOF particles can entrap the AuBPs and finer particles are less likely to cover the NPs. Do you have TEM images to confirm that the AuBPs are trapped inside the larger

MOFs (at high temperature) and are not trapped inside the finer MOFs (at lower temperature).

Response 2-5: We extend our gratitude to reviewer #2 for the valuable comment regarding the entrapment of AuBPs within the MOF during the photothermal synthesis. This input has highlighted the need for further investigation of the AuBP-MOF composite's formation mechanism. We have now conducted additional experiments with the aim of deepening our understanding of the composite material's structure and the nature of the interaction between the MOF and AuBPs. This concern was also raised by reviewer # 1, thus, we kindly direct the reviewer to **Response 1-3** where the proceedings of these experiments are elaborated in detail.

We sincerely appreciate this contribution, which has deepened our understanding of the mechanisms underlying our photothermal synthesis approach.

6. How does the size of MOFs compare to AuBPs? Please provide a size distribution profile of MOFs formed at different temperature in complement with the SEM images to facilitate the size comparison. In the case of 60°C, if the MOF size is larger than the AuBPs, can you quantify the amount of AuBPs being trapped inside the MOF?

Response 2-6: We appreciate reviewer # 2's question regarding the size comparison between MOFs and AuBPs. As mentioned in Response 1-3, our investigations have revealed that the attachment of zirconium oxide to the silica shell during the heating process promotes the formation of MOF on the AuBPs, and the influence of the size of the MOF particles on this phenomenon appears to be less dominant.

Nonetheless, we have conducted SEM imaging of the photothermally synthesized AuBP@UIO-66 composites to facilitate the size comparison and included a size distribution profile of the UIO-66 particles synthesized at different temperatures (Supplementary Fig. 31-33). However, measuring the precise sizes of the UIO-66 particles is challenging due to their undefined shape. This may explain why the sizes of the UIO-66 particles synthesized at 80 °C and 100 °C appear to be similar. In any case, the MOF particles are consistently smaller than an AuBP@SiO₂ in all synthesis temperatures.

Importantly we observed a more significant increase in the size of the MOF aggregates, which are structures composed of multiple units of the MOF particles, as the synthesis temperature is increased. These aggregates are larger than AuBPs and may contribute to the entrapment of AuBPs within the MOF.

Regarding the quantification of AuBPs trapped inside the MOF at 60 °C, we have provided the amount of gold for each synthesis condition (60-100 °C) in the ICP-OES results presented in Figure 2e of the manuscript.

7. In part 3 (Versatility and scope of the photothermal synthesis), please include the reference on PXRD and SEM of the synthesized MOFs to support the experimental PXRD and SEM obtained.

Response 2-7: We appreciate reviewer #2 for bringing to our attention the missing references regarding the PXRD and SEM characterization of the synthesized MOFs. We

apologize for the oversight and have now included the appropriate references in the manuscript (61-63) to support the experimental PXRD and SEM results obtained.

8. The formation of HKUST-1 in part 3 took place in EtOH/H₂O 1:1 solvent. How does the authors reach temperature of 120oC while the solvent boiling point is around 80-90oC?

Response 2-8: To achieve a temperature higher than the boiling point of the solvent, we utilized a standard glass pressure tube. This setup allowed us to create a controlled environment where the temperature could be raised to 120 °C while using the EtOH/H₂O 1:1 solvent. We apologize for not mentioning this detail in the manuscript and appreciate the reviewer for pointing it out.

9. Similarly, in part 4 (AuBP-embedded plasmonic MOF), line 182, the plasmonic UiO-66 reached 250oC, although DMF boiling point is 153oC. Please explain.

Response 2-9: As the reviewer rightly points out the MOF was synthesized in DMF, however, the mentioned experiments were carried out with the isolated product, without any solvent (Fig. 4b). We have now revised the manuscript to clarify this point.

10. In part 5 (Exploring applications of the photothermal MOF), the mechanism for acceleration of water/solvent desorption remains unclear. Both methods rely on heat-induced evaporation of water/solvent, hence the rate of desorption should be limited to the evaporation of water/solvent molecules. If the higher temperature at the hotspot cause the acceleration of evaporation/desorption, it should also be possible to achieve similar rate of desorption by increasing the temperature of the conventional heating process. Please elaborate on how the generation of hotspot can accelerate the water/solvent desorption.

Response 2-10: We appreciate the insightful comment from reviewer #2 regarding the mechanism of water/solvent desorption in the photothermal MOF. As we believe, the generation of hotspots near the AuBPs during irradiation plays a role in accelerating the desorption process. As described in the manuscript, Although the bulk temperature might not reach extreme levels, the immediate vicinity of the AuBPs reaches significantly higher temperatures, potentially leading to accelerated desorption rates.

Inspired by the suggestion of reviewer #2, we conducted additional experiments involving the heating of UiO-66 with water at different temperatures, including 100 °C and 120 °C, as described in part 5 (Exploring applications of the photothermal MOF) of the manuscript. Surprisingly, the water completely desorbed in 5 minutes at 100 °C and 2 minutes at 120 °C. In comparison, as shown in the manuscript, the photothermal method achieved complete desorption in just 1 minute at 85 °C. These findings, where the desorption rate at much elevated temperatures is comparable to the photothermal desorption, support the assumption that the generated hotspots in the photothermal method may significantly enhance the desorption rate compared to conventional heating at higher temperatures. We have included these new results in supplementary figure 76 to provide additional evidence for the faster desorption achieved through the photothermal method, due to the generation of hotspots.

Reviewer #3 (Remarks to the Author):

in this manuscript Weizmann et al. report the light induced MOF synthesis due to the photothermal effect of gold nanoparticles and the resulting temperature increase on the reaction media. the resulting composite show also photothermal effect and this effect can be applied for the fast activation of MOFs. first i would like to highlight that some important ref. on the photothermal effect on MOFs are missing. especially the one from espin et al. (10.1021/acsami.8b00557) is important because the photothermal effect (using UV light) was used for the fast activation of different MOFs such as HKUST-1, Zlf-67, Uio-66-X, etc... the same advantage that authors claim in this manuscript. there are also others...

Response 3- general summary: We would like to thank reviewer #3 for thoroughly reviewing our manuscript. We appreciate the referee bringing to our attention that we were missing important references on the photothermal activation of MOFs, and specifically the work by Espín *et al.*. This reference is indeed very relevant as it highlights the use of UV and visible light for photothermal activation of various MOFs. We have revised our manuscript according to this feedback and incorporated relevant references, including the study of Espín *et al.* (References 49-50, 54)

the in situ synthesis of MOFs thank to the heat generated by gold nanoparticles is interesting but i would like to know how the resulting composite is... how the gold is distributed in the MOF. Microscopy is not very clear about this.

Response 3-1: We appreciate the valuable comment from reviewer #3. We acknowledge that our manuscript lacked clarity in characterization of the resulting composite in our study. Through extensive investigation using TEM, XPS, EDS and ICP-OES analysis, we discovered that amorphous zirconium oxide layer becomes attached to the silica shell of the gold nanoparticles during the heating process. This attachment facilitates the growth of the MOF around the nanoparticles. We have provided detailed results and further information on this phenomenon in comment 1.3.

the other question that i have is about the need of this King of synthesis to have this fast activation, did the authors try to mix the nanoparticles with already made MOF particles and activate them?

Response 3-2: We appreciate the insightful question from reviewer #3 regarding the possibility of activating MOFs by mixing them with AuBPs and subjecting them to photothermal treatment. To address this, we conducted an experiment where we synthesized UIO-66 using conventional methods. We then added AuBPs to the MOF particles in equal amounts (5 OD) and physically mixed them together. Subsequently, we subjected the mixture to photothermal activation for 5 minutes, following the same procedure as described in our research.

Interestingly, the BET surface area measurement of the resulting composite showed a lower value of 990 m²/g compared to the surface area of pure UIO-66. This suggests that

the activation of the MOF was not fully achieved using this method. We hypothesize that during the drying process, the AuBPs tended to aggregate, resulting in an uneven distribution within the MOF matrix and less effective activation. This highlights the importance of the photothermal synthesis method in achieving a more efficient composite for photothermal activation.

It is important to note that we only conducted the activation for a short duration of 5 minutes, and further activation may have yielded better results. Therefore, this method still holds potential for efficient activation with a slightly longer duration.

This experimental procedure requires further investigation, and we appreciate reviewer #3 for providing us with the idea. We will continue to explore this approach in our future research to optimize the activation process.

did the authors try to synthesis core Shell particles with this approach?

Response 3-3: We appreciate the insightful question from reviewer #3 regarding the synthesis of core-shell particles using our photothermal approach. Our findings indicate that the size of the MOF particles in our study is not significantly smaller than that of the AuBPs, which may present a challenge for achieving uniform distribution around the silica shell. In an attempt to increase the affinity of AuBPs to UIO-66, we heated the AuBPs in the presence of $ZrCl_4$ at 100 °C for one hour. Subsequently, we introduced the zirconium-coated AuBPs into the UIO-66 precursors solution and synthesized the MOF photothermally. Interestingly, TEM analysis revealed AuBPs attached to small UIO-66 particles (revision fig. 4). While these observations are intriguing, they also underscore the difficulty in attaining core-shell particles under these specific conditions, particularly due to the similar sizes of the particles and the aggregating nature of the MOF. Thus, achieving consistent core-shell structures would necessitate further research.

Additionally, inspired by reviewer #3's comment, we explored the literature and came across two interesting studies that describe the synthesis of core-shell particles consisting of UIO-66 covering silica particles.^{7,8} In both cases, the researchers used specific reagents, such as amine groups and carboxylic acid groups, to facilitate the attachment of UIO-66 onto the silica surface. Drawing insights from these references and our own observations, we intend to incorporate these approaches into our future research endeavors.

We would like to thank reviewer #3, based on this comment and the insights mentioned above, we are encouraged to explore the possibility of synthesizing core-shell particles using a photothermal method in our future research. Unfortunately, this exploration falls beyond the scope of our current research, and therefore, will not be included in the revised manuscript.

Revision Figure 4 | TEM images of UiO-66 synthesized with zirconium coated AuBPs using 850 nm 100 W LED at 100 °C for 20 minutes.

to conclude I have my doubts about the novelty (specially considering this previous study on fast photothermal activation) and I would like to see a reviewed manuscript before to consider the acceptance of this manuscript in nature communications

Response 3-4: We appreciate the valuable feedback and concerns raised by reviewer #3 regarding the novelty of our research. In response, we have thoroughly revised our manuscript, and incorporated the reviewer's suggestions, providing additional explanations to strengthen the novelty and significance of our work.

Our revised manuscript now presents a clearer and more comprehensive account of our research, emphasizing the distinctive features of our photothermal synthesis of MOFs, which stands apart from previous studies on fast photothermal activation.

We hope that our detailed response has adequately clarified the developments our approach brings to the field.

In comparing our study to Espín *et al.*, it is important to note that while both methods involve light-induced heating and demonstrate similar advantages, there are fundamental differences. The photothermal activation in Espín *et al.*'s work relies on the heat generated when the MOF matrix itself absorbs light. This method's effectiveness depends on the specific MOF used, and as described in their paper, not all MOFs reach high temperatures (e.g., UiO-66 only reached 57 °C). In contrast, our photothermal activation method does not depend solely on the MOF itself. We introduce a photothermal agent into the reaction mixture, allowing us to control the temperature the MOF reaches by adjusting the concentration of the photothermal material. Another distinction is that Espín *et al.* focused on UV-vis light activation, while our method covers a wider range of the spectrum, from UV to IR, independent of the MOF structure.

It is crucial to emphasize that the core focus of our research is the development of photothermal synthesis of MOFs, with photothermal activation being just one of the benefits derived from our novel approach.

We are confident that these revisions have addressed the reviewer's concerns and further strengthened the merit of our work for consideration in *Nature Communications*.

Response references:

1. Zhang, H., Zhu, T. & Li, M. Quantitative Analysis of the Shape Effect of Thermoplasmonics in Gold Nanostructures. *J. Phys. Chem. Lett.* **14**, 3853–3860 (2023).
2. Jain, P. K., Lee, K. S., El-Sayed, I. H. & El-Sayed, M. A. Calculated absorption and scattering properties of gold nanoparticles of different size, shape, and composition: Applications in biological imaging and biomedicine. *J. Phys. Chem. B* **110**, 7238–7248 (2006).
3. Yang, W. *et al.* Precise control over the silica shell thickness and finding the optimal thickness for the peak heat diffusion property of AuNR@SiO₂. *J. Mater. Chem. B* **10**, 364–372 (2022).
4. Katz, M. J. *et al.* A facile synthesis of UiO-66, UiO-67 and their derivatives. *Chem. Commun.* **49**, 9449–9451 (2013).
5. Jiang, R., Cheng, S., Shao, L., Ruan, Q. & Wang, J. Mass-based photothermal comparison among gold nanocrystals, PbS nanocrystals, organic dyes, and carbon black. *J. Phys. Chem. C* **117**, 8909–8915 (2013).
6. Han, H. S. & Choi, K. Y. Advances in nanomaterial-mediated photothermal cancer therapies: Toward clinical applications. *Biomedicines* **9**, 1–15 (2021).
7. Zhang, X., Han, Q. & Ding, M. One-pot synthesis of UiO-66@SiO₂ shell-core microspheres as stationary phase for high performance liquid chromatography. *RSC Adv.* **5**, 1043–1050 (2015).
8. Arrua, R. D. *et al.* UiO-66@SiO₂ core-shell microparticles as stationary phases for the separation of small organic molecules. *Analyst* **142**, 517–524 (2017).

REVIEWER COMMENTS

Reviewer #1 (Remarks to the Author):

After reading the revised version of manuscript as well as the response letter, I think that the authors did not fully address my concerns. In general, I still do not think that the thermoplasmonic properties of AuBP@SiO₂ provides new scientific insights to justify its acceptance in Nature Communications, that is, I found a lack of noteworthy results. Specially comparing with other absorbing materials and even the temperature synthesis. From the results, the photothermal synthesis need to be performed at 60°C but in the temperature synthesis, it can be developed at 100°C in high yields. The absence of a proper scale up is also a drawback.

Reviewer #3 (Remarks to the Author):

I believe the authors have put significant effort into improving the paper by including more characterization and experiments. Therefore, I recommend the publication of this manuscript.

We would like to express our sincere gratitude to the reviewers for dedicating their time and effort to review our manuscript once again. We hope that this letter will address all your concerns and misunderstandings.

Reviewers' comments:

Reviewer #1 (Remarks to the Author):

After reading the revised version of manuscript as well as the response letter, I think that the authors did not fully address my concerns. In general, I still do not think that the thermoplasmonic properties of AuBP@SiO₂ provides new scientific insights to justify its acceptance in Nature Communications, that is, I found a lack of noteworthy results. Specially comparing with other absorbing materials and even the temperature synthesis.

We greatly appreciate the time and effort dedicated by reviewer #1 to conduct a re-evaluation of our manuscript. We are genuinely concerned to learn that the reviewer's reservations persist. Allow us to clarify the focal point of our research. While our work delves into the thermoplasmonic properties of AuBPs, it's crucial to emphasize that this aspect serves as an instrumental tool rather than the central theme of our study. Our manuscript introduces a concept: the photothermal synthesis of MOFs. Thus, we agree with the statement that thermoplasmonics of AuBPs didn't provide new scientific insight, because it shouldn't, the photothermal synthesis of MOF and the formation of a photothermal composite provides the scientific insights. The choice of AuBPs as our primary photothermal agents was influenced by their advantages, notably high-yield synthesis and monodispersity. However, as we have elaborated in our revised manuscript, the method can be easily adapted to various other photothermal reagents, rendering it a versatile and user-friendly approach. Thus, comparing the use of AuBPs to other absorbing materials becomes less relevant, as the diversity of suitable reagents becomes an asset of our synthesis method. Furthermore, our research showcases the significant benefits of photothermal synthesis. It not only accelerates the process, making it energy-efficient, but also opens up new possibilities for creating composite materials, forming a central contribution to the field. We hope these clarifications help highlight the significance and innovation of our work.

From the results, the photothermal synthesis need to be performed at 60°C but in the temperature synthesis, it can be developed at 100°C in high yields.

In our manuscript, we applied the photothermal synthesis not only at 60 °C but also at 80 °C and 100 °C. The characterizations we performed, including PXRD, BET, SEM, and TEM analyses, consistently confirmed that the photothermal synthesis at all three temperatures resulted in the desired MOF with a high surface area. The only difference

between these syntheses was the amount of AuBPs that remained inside the MOF, which we observed to vary. In comparison to conventional synthesis, our photothermal method exhibited not only equivalent yields but also considerably shorter reaction durations. Furthermore, the synthesis relies on a simple, low-energy consumption LED, making it easy to activate and monitor. We believe that these insights strengthen the scientific value of our research and showcase the versatility and efficiency of photothermal MOF synthesis.

The absence of a proper scale up is also a drawback.

Scaling up a laboratory synthesis to a liter-scale is undoubtedly a significant step, often involving complex engineering challenges and considerable resources. In our work, we executed our photothermal synthesis of MOF at different volumes, ranging from 1 ml to 20 ml. In our initial submission, we referred to the 20 ml synthesis as a "scale-up." However, we have addressed the reviewer's concern by more accurately describing it as "increased volume". For a full-scale industrial-level implementation, adapting our system to large-scale production would indeed be a substantial endeavor. This kind of scale-up typically involves designing and building entirely new equipment, a process that would require a significant amount of time and resources. It is also more suited to an industrial setting than a typical scientific laboratory. While we have not demonstrated a liter-scale synthesis, we are confident that, with optimization, scaling up the photothermal synthesis is feasible. With the right arrangement of LEDs, we believe it can remain energy-efficient and rapid, thus making it a suitable method for larger-scale production. Importantly, the scientific significance of our work lies in pioneering this synthesis method, offering a promising avenue for further research and potential applications. Although we haven't showcased a large-scale synthesis, other research groups can certainly use our method in milliliter-scale production, which provides a sufficient quantity of product for further scientific investigations.

Reviewer #3 (Remarks to the Author):

I believe the authors have put significant effort into improving the paper by including more characterization and experiments. Therefore, I recommend the publication of this manuscript.

We extend our sincere gratitude to reviewer #3 for the time and effort devoted to the evaluation of our manuscript. The constructive feedback and recommendations have been invaluable in enhancing the quality and comprehensiveness of our work. We're pleased to hear that the additional characterizations and experiments we incorporated

have met your expectations. The recommendation for the publication of our manuscript is both encouraging and greatly appreciated.